# Interestingness First Classifiers

## Abstract

Most machine learning models are designed to maximize predictive accuracy. In this work, we explore a different goal: building classifiers that are interesting. An "interesting classifier" is one that uses unusual or unexpected features, even if its accuracy is lower than the best possible model. For example, predicting room congestion from CO2 levels achieves near-perfect accuracy but is unsurprising. In contrast, predicting room congestion from humidity is less accurate yet more nuanced and intriguing. We introduce EUREKA, a simple framework that selects features according to their perceived interestingness. Our method leverages large language models to rank features by their interestingness and then builds interpretable classifiers using only the selected interesting features. Across several benchmark datasets, EUREKA consistently identifies features that are non-obvious yet still predictive. For example, in the Occupancy Detection dataset, our method favors humidity over CO2 levels and light intensity, producing classifiers that achieve meaningful accuracy while offering insights. In the Twin Papers dataset, our method discovers the rule that papers with a colon in the title are more likely to be cited in the future. We argue that such models can support new ways of knowledge discovery and communication, especially in settings where moderate accuracy is sufficient but novelty and interpretability are valued.

## 1 Introduction

Machine learning research usually focuses on building models with the highest possible accuracy. This is natural, since accurate predictions are useful for applications such as medical diagnosis [10, 13, 14], recommender systems [16, 24, 25], or speech recognition [19, 42, 43].

However, accuracy is not the only possible goal. Since data involve aleatoric uncertainty [20, 21], perfect accuracy may be out of reach. There may not exist any "true" laws waiting to be discovered. This leads to our motivation: faithfulness should not be regarded as an absolute objective, and even rules that are not entirely faithful may still be worth pursuing if they offer interesting insights.

For illustration, consider the problem of classifying whether a user is an adult based on their profile. If the profile contains an `age` feature, then the simple classification rule `age` $\geq 18$ achieves 100% accuracy. On social networking services, users may occasionally misreport their age, but in most cases they provide approximately correct information, so an accuracy of around 99% is still realistic. This rule is extremely accurate, but it is not interesting.

Let us consider a more unusual rule. Suppose we define the classifier: "if the registered email address ends with @icloud.com then the user is a minor, whereas the user is an adult if it ends with @aol.com." Such a rule might achieve some level of accuracy. Although its accuracy would not be particularly high, it is engaging precisely because it is unexpected. This is the kind of classifier we would like to obtain.

A somewhat more academic example is the detection of depressive symptoms. Numerous tests have been proposed for this task. A well-known one is the Beck Depression Inventory, which asks participants to answer 21 questions on a four-point scale—such as whether they feel depressed, whether they are pessimistic about the future, or whether they feel guilty—and judges depression based on the total score. This test is widely used for diagnosing depression, but whether the decision rule itself is "interesting" is questionable. By contrast, a more intriguing rule is known [46]: "if a person uses first-person pronouns (I, my, me, myself)

frequently in an essay, then they are likely to be depressed." Those with depressive tendencies are believed to pay greater attention to information about themselves than to information about the outside world or people around them. Of course, this rule is far from perfect: it is heavily influenced by an individual's writing style, and even cheerful people may frequently use first-person pronouns. In fact, while the rule is statistically significant, its predictive power is quite low (around r = 0.13) [9]. Nevertheless, it is considered interesting, and despite its limited accuracy, it has gained recognition in psychology, inspiring a range of related and follow-up studies [9].

As a more practical example used in this paper, consider the task of detecting whether an office is occupied (the Occupancy Detection dataset). In this dataset, a simple rule based on whether the lights are on achieves around 99% accuracy. Offices are typically lit during working hours, and in most companies people do not remain in an office with the lights off, so the relation "lights on = people present" holds almost universally. However, if the problem can be solved in this trivial way, it is not very interesting. As we will show later, the same problem can also be addressed with the rule "if humidity is high, then people are present," which achieves around 85% accuracy. Of course, high humidity alone cannot distinguish between rainy weather and actual human presence, and even when people are present the humidity does not always rise significantly, so the accuracy cannot be perfect. Nevertheless, the idea that occupancy can be inferred from a humidity sensor sits at an intriguing middle ground—it feels like it could work, but also like it might not, making it nuanced and interesting. Moreover, beyond mere curiosity, the fact that non-obvious sensors can partially solve the task suggests practical potential: applications that do not require extremely high accuracy might be realized at lower cost than conventional approaches.

Related topics include feature selection, interpretability, and explainability. However, these are generally aimed at maximizing accuracy, which distinguishes them from our focus on interestingness. For example, in the adulthood classification problem, feature selection would naturally identify `age` as the most important feature, leading to the interpretable rule `age` $\geq 18$. This rule is interpretable and highly accurate, but not interesting. In the office occupancy detection problem, the feature `light` would be selected, producing the rule `light` $\geq 1$. Again, this is interpretable and accurate, but not interesting.

In this paper, we introduce EUREKA (Exploring Unexpected Rules for Expanding Knowledge boundAries), a simple framework for building interesting classifiers. The method uses large language models (LLMs) to compare pairs of features and judge which one would produce a more interesting rule. From these pairwise comparisons, we derive a global ranking of features and train interpretable classifiers using only the top-ranked ones.

Our experiments across multiple benchmark datasets show that EUREKA consistently highlights features that are not the most predictive but still achieve non-trivial accuracy. For instance, instead of selecting light intensity to predict office occupancy, which is highly predictive but obvious, EUREKA automatically prefers humidity, which provides a weaker but more surprising signal.

The main contributions of this work are:

- We define the concept of interesting classifiers, which prioritize unexpectedness over accuracy.

- We propose EUREKA, a simple framework that leverages LLM-based pairwise comparison to rank features by interestingness.

- We demonstrate, through experiments on several datasets, that classifiers built from top-ranked interesting features can still reach meaningful accuracy while providing surprising and interpretable insights.

**Reproducibility**: Our code is available at `https://github.com/xxxxxx/eureka` (to be filled in the camera-ready).

## 2 Proposed Method

### 2.1 Problem Setting

We consider a standard classification problem on tabular data. The input is a training dataset $\mathcal{D} = \{(\boldsymbol{x}_i, y_i) \mid i = 1, \ldots, n\} \subset \mathbb{R}^d \times \mathcal{Y}$, where $\boldsymbol{x}_i \in \mathbb{R}^d$ is the feature vector, and $y_i \in \mathcal{Y}$ is the label.

For example, in the case of Occupancy Detection, the dataset might include features such as `Temperature`, `Humidity`, `Light`, `CO2` levels, and `HumidityRatio`, with the label indicating whether the room is occupied, as follows:

| id | Temperature | Humidity | Light | CO2 | HumidityRatio | Occupancy $(y)$ |
|----|-------------|----------|-------|-------|---------------|-----------------|
| 1  | 23.18       | 27.27    | 426.0 | 721.3 | 0.004793      | 1               |
| 2  | 23.15       | 27.27    | 429.5 | 714.0 | 0.004783      | 1               |
| 3  | 23.15       | 27.25    | 426.0 | 713.5 | 0.004779      | 1               |
| 4  | 23.15       | 27.20    | 426.0 | 708.3 | 0.004772      | 1               |
| 5  | 23.10       | 27.20    | 426.0 | 704.5 | 0.004757      | 1               |
|    |             |          | . . . |       |               |                 |

The output is a classifier. It is important to note that the input-output setting of our problem is exactly the same as that of conventional classification tasks, namely tabular data as input and a classifier as output. The key distinction lies in our objective: instead of maximizing accuracy, we focus on the interestingness of the resulting classifier.

### 2.2 Overview of EUREKA

Our framework, named **EUREKA** (Exploring Unexpected Rules for Expanding Knowledge boundAries), consists of three main components:

1. **Interestingness ranking:** Rank all features by interestingness using LLM-based pairwise comparisons.

2. **Classifier construction:** Train an interpretable model using only the top-$K$ features in the ranking.

3. **Interestingness-first selection:** Increase $K$ gradually until it becomes predictive enough.

### 2.3 Pairwise Ranking by LLMs

To assess interestingness, we ask a large language model to compare two candidate features $f_i$ and $f_j$: "If one could predict the label (e.g., Room Occupancy) only from $f_i$ (e.g., Humidity) and only from $f_j$ (e.g., CO2), which prediction rule would be more interesting?" From these pairwise judgments, we derive a global ranking using a pairwise Borda count [49]: each feature receives one vote when it is selected in a comparison, and features are ordered by their total votes. Although this procedure is simple, it has been shown to be, up to constant factors, an information-theoretically optimal procedure [49, Theorem 2].

For example, when we apply this procedure to the classification problem of determining whether an office is occupied (the Occupancy Detection dataset), we obtain the following ranking of features:

1. `HumidityRatio` (the ratio of the mass of water vapor to the mass of dry air)
2. `Humidity`
3. `Temperature`
4. `Light`
5. `CO2`

This means that it is interesting if occupancy can be predicted solely from the amount of water vapor, whereas it is unsurprising if occupancy can be predicted solely from CO2 levels.

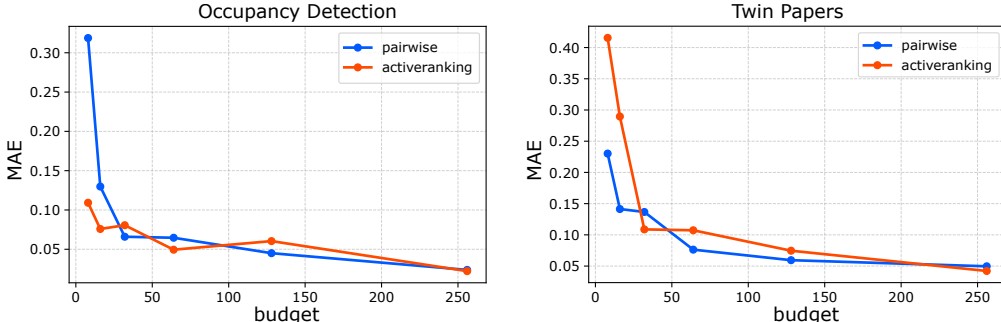

Figure 1: **Comparison of ranking methods.** Mean absolute error (MAE) of estimated Borda scores against the ground truth, as a function of the number of pairwise comparisons $N$. Both methods achieve similar accuracy. Lower is better.

## 2.4 Comparison with Active Ranking

An alternative is *active ranking from pairwise comparison* [17], which adaptively selects which items to compare based on confidence bounds, similar to successive elimination [41]. In theory, this method can achieve almost optimal sample efficiency up to logarithmic factors [17, Theorem 1]. However, in practice we found it overly conservative: many pairs remain unresolved, and the algorithm does not significantly reduce the number of queries. Both pairwise counting and active ranking estimate the Borda score

$$b_i = \frac{1}{n-1} \sum_{j \neq i} \Pr[i \text{ wins } j] \tag{1}$$

and rank items accordingly. We treat the $b$ estimated by the Monte Carlo sampling with $N = 4096$ comparisons as the (surrogate) ground truth, and then restrict the number of comparisons to $N = 8, 16, 32, 64, 128, 256$ to compute the estimates $\hat{b}$ with each method. We evaluate the mean absolute error (MAE) of $\hat{b}$ against the ground truth. Experiments are conducted on the *Occupancy Detection* and *Twin Papers* datasets [47]. The results are plotted in Figure 1. We observed almost no difference in accuracy between active ranking and pairwise counting. Furthermore, active ranking is difficult to parallelize due to its sequential and active nature, while pairwise counting can be evaluated in parallel, leading to faster wall-clock time. For our purposes, this practical advantage is important.

## 2.5 Direct Ranking vs. Pairwise Ranking

We also tested asking an LLM to directly produce a full ranking of features. However, this approach was unstable: repeated runs produced different orders. To quantify this, we generated 20 rankings and measured rank correlations between them. Table 1 shows that pairwise ranking is significantly more stable than direct ranking. Although these datasets contain only 5 and 7 features, respectively, the direct output method still pro-

| Dataset | Pairwise | Direct |
|---|---|---|
| Occupancy (Kendall $\tau$) | **0.854 ± 0.128** | 0.615 ± 0.369 |
| Occupancy (Spearman $\rho$) | **0.916 ± 0.088** | 0.707 ± 0.337 |
| Twin Papers (Kendall $\tau$) | **0.766 ± 0.135** | 0.689 ± 0.344 |
| Twin Papers (Spearman $\rho$) | **0.876 ± 0.096** | 0.763 ± 0.334 |

Table 1: Average correlations between rankings generated across 20 runs. Pairwise counting consistently outperforms direct ranking in stability. Higher is more stable.

duces unstable results. As the number of features grows, it becomes even harder for direct output to retain context, leading to greater instability. In contrast, pairwise comparison always requires evaluating only two features at a time, making it relatively more stable.

Direct ranking has the advantage of fewer API calls, so in low-budget scenarios it may still be attractive. However, in this paper we mainly consider the pairwise method.

## 2.6 Interestingness-First Classifiers

Once we obtain a ranking of features, we construct classifiers with only the top-$K$ features. Formally, we solve

$$\max_{\theta} \quad \mathrm{Acc}(\theta; X_{:,S_K}, y) \quad \text{s.t. } S_K = \{\text{top } K \text{ features by interestingness}\},$$

where $\theta$ are model parameters, and $K$ is a hyperparameter.

The key constraint is that we may *only* use the top-$K$ interesting features, regardless of how predictive the other features may be. The ordering of features is guided exclusively by interestingness, which is why we refer to the approach as interestingness-first. In practice, we start with $K = 1$. If the resulting classifier is not predictive enough, we increase to $K = 2$, and so on. Let $K'$ denote the smallest such value. Then: Features ranked $1, \ldots, K' - 1$ are interesting but not predictive alone. The set $S_{K'}$ yields the first classifier that is both interesting and predictive. This feature set is the most interesting among predictive sets, in the sense that its least interesting feature is more interesting than the least interesting feature in any other set.

Note that features with no predictive power at all may sometimes be selected in the top-$K$ features. This occurs because features that appear uninformative at first glance are often perceived as more interesting, and thus tend to be ranked highly—even though, unfortunately, they truly lack predictive ability as expected in many cases. If such a feature unexpectedly turns out to have predictive power, it is a fortunate bonus rather than a usual consequence. Thus, some of the top-$K$ candidates may be non-predictive features, but they are handled by subsequent models. In practice, when the model is trained, such features are largely ignored. If necessary, sparse regularization such as $L_1$ can be applied to ensure that these uninformative features are discarded, leaving only those that are both interesting and predictive.

## 2.7 Choice of Base Classifier

Finally, it is important that the classifier itself be interpretable, otherwise the "interesting rule" cannot be communicated clearly. In other words, interpretability is a necessary (though not sufficient) condition. While many complex models could be trained, we restrict ourselves to inherently interpretable methods such as logistic regression or shallow decision trees. In this paper, we use logistic regression by default.

# 3 Experiments

In this section, we evaluate EUREKA across six representative classification tasks. Our goal is not to maximize accuracy. The resulting classifiers may sometimes outperform the chance rate only slightly. However, we do not regard low accuracy as problematic; as long as the performance exceeds the chance rate, we prioritize evaluating the rule for its interestingness.

## 3.1 Setup

**Tasks and datasets.** We consider six tabular classification problems where both obvious and non-obvious explanatory rules exist:

1. **Occupancy Detection**: predict whether an office room is occupied.

2. **Twin Papers**: given a pair of similar academic papers, predict which one will be cited more often.

3. **Mammographic Mass**: classify mammographic masses as benign or malignant.

4. **Breast Cancer Wisconsin**: classify masses as benign or malignant.

5. **Adult**: predict whether annual income exceeds $50k.

6. **Website Phishing**: distinguish phishing sites from benign ones.

These span physical sensing, scientometrics, medical diagnostics, and security domains, providing diverse ground for "interesting" signals.

Table 2: **Top features selected by each method.**

| Dataset | Task description | Group LASSO | Logistic Regression | Validation Selection | EUREKA (ours) |
|---|---|---|---|---|---|
| Occupancy Detection | Predict whether a room is occupied | `Light` | `Light` | `Light` | `HumidityRatio, Humidity` |
| Twin Papers | Predict which paper is cited more | `Lengthen_the_ reference` if the reference list is longer | `Lengthen_the_ reference` if the reference list is longer | `Lengthen_the_ reference` if the reference list is longer | `Including_a_ Colon_in_the_Title` |
| Mammographic Mass | Predict benign vs. malignant mass | `BI-RADS` score assigned by radiologist | `BI-RADS` score assigned by radiologist | `BI-RADS` score assigned by radiologist | `Density, Age` mass density and patient age |
| Breast Cancer Wisconsin | Predict benign vs. malignant tumor | `Bare_nuclei` frequency of exposed nuclei | `Bare_nuclei` frequency of exposed nuclei | `Uniformity_of_ cell_size` | `Marginal_adhesion` cell-to-cell adhesion strength |
| Adult | Predict whether income > \$50k | `capital-gain` | `capital-gain` | `capital-gain` | `capital-loss` |
| Website Phishing | Predict whether a site is phishing | `SFH (Server Form Handler)` | `SFH (Server Form Handler)` | `SFH (Server Form Handler)` | `popUpWindow` |

**Detailed Settings.** We adopt a fixed pipeline across tasks: numeric features are standardized, categorical features one-hot encoded, and missing values imputed based on the statistics of the training data (median for numerics, mode for categoricals). Datasets are stratified into train/test splits (80/20). We use gpt-5-nano as our language model throughout the experiments. The chance rate refers to the accuracy obtained when always predicting the majority class. To ensure that the fitted models are non-trivial, we also report a likelihood-ratio test against an intercept-only null model. The significance level is set at $\alpha = 0.05$ with Bonferroni correction.

## 3.2 Selected Features

We compare the features selected by EUREKA and conventional feature selection methods:

- **Group LASSO** [32, 65]: Since categorical features are one-hot encoded, a single feature may span multiple dimensions. We therefore employ Group LASSO rather than vanilla LASSO.

- **Logistic Regression**: We use $L_2$ regularization and select features with large weights. When a feature spans multiple dimensions, we treat its weights as a vector and use the norm as its score.

- **Validation-based selection**: We split the training data into validation sets. For each feature, we train a single-feature logistic regression and select the features that achieve high accuracy on the validation set.

Table 2 shows the top features selected by each method. For the proposed method, it sometimes happens that a feature with no predictive power at all is chosen first; in such cases, we also report the second-ranked feature. This never occurs with conventional methods, since they always prioritize predictive strength. Such outcomes arise precisely because our method prioritizes interestingness over predictive power.

Although the baselines (Group LASSO, Logistic, Validation Selection) differ in their criteria, they all converge on features that are highly predictive to maximize accuracy. As a result, they tend to select almost the

same variables across datasets. In contrast, EUREKA produces quite different feature choices, reflecting its prioritization of interestingness over raw predictiveness.

As noted earlier, in the Occupancy Detection dataset, which aims to predict whether a room is occupied, the most predictive feature is indoor light intensity. However, when prioritizing interestingness, humidity emerges as the selected feature.

In the Twin Papers dataset, where the task is to predict which of two similar papers will be cited more often, the most predictive feature is whether the reference list is long. This is quite reasonable: a long reference list suggests that the paper is thoroughly grounded in prior research, which increases its chances of being high quality and therefore highly cited. Moreover, in modern settings, citations appear on platforms such as Google Scholar as backlinks, meaning that including more references can also increase visibility and exposure, thereby creating further opportunities to be cited.

By contrast, when interestingness is prioritized, the selected feature becomes whether the paper's title contains a colon. Although this may seem like a rather peculiar feature, we will later see that it can predict citations to a meaningful degree.

For the Mammographic Mass dataset, which concerns predicting whether a breast mass is benign or malignant, the BI-RADS score is a key feature. This score is assigned manually by radiologists, making it highly reliable as it reflects expert clinical judgment. Because it is so strong a predictor, competitions and benchmarks often exclude it to avoid making the task trivial. In this study, however, we intentionally included it in the experiments. Naturally, conventional feature selection methods identified BI-RADS as the most predictive feature. By contrast, our proposed method regards relying on BI-RADS as uninteresting, and instead selects other features such as Density or Age even in the presence of BI-RADS.

In the Adult dataset, which predicts whether a person earns more than \$50,000 per year, conventional methods select capital-gain as the most predictive feature, whereas our method selects capital-loss. Indeed, losses can also be informative for predicting income, but this is more nuanced and interesting than simply focusing on gains.

Interestingly, when logistic regression is trained using capital-loss, the resulting model shows that the greater the losses, the higher the probability of earning more than \$50,000. This rule yields reasonably good predictive accuracy. While the idea that larger losses imply higher income may appear counterintuitive at first, one plausible explanation is that individuals earning less than \$50,000 may lack significant assets to sell in the first place, whereas those with substantial capital losses are likely to own considerable assets—such as in real estate transactions—making them more likely to belong to the high-income group. This is, in itself, an interesting finding.

### 3.3 Accuracy Using Interesting Features Only

Next, we train classifiers using the features selected by the proposed method. Figure 2 presents the accuracy obtained for different values of $K$.

In most cases, using only the single top-ranked interesting feature already achieves accuracy above the chance rate (the accuracy of always predicting the majority class). There are two exceptions: (*i*) the `HumidityRatio` feature in Occupancy Detection, and (*ii*) the `Density` feature in Mammographic Mass. For Occupancy Detection, `HumidityRatio` is statistically significant, but its effect size is small and the dataset is class-imbalanced; as a result, its test accuracy does not reach the chance baseline. Similarly, `Density` alone is insufficient in Mammographic Mass. In both cases, however, adding the second-ranked interesting feature (e.g., `Humidity` for Occupancy; `Age` for Mammographic Mass) is enough to surpass the chance rate.

As mentioned earlier, in Occupancy Detection this yields the rule "higher humidity $\Rightarrow$ occupied," which reaches about 85% test accuracy—a simple and communicable alternative to the trivial light-based rule.

In Twin Papers, using only `Including_a_Colon_in_the_Title` achieves about 52% accuracy. This corresponds to titles such as "*Twin Papers: A Simple Framework of Causal Inference for Citations via Coupling*" or "*Beyond Exponential Graph: Communication-Efficient Topologies for Decentralized Learning via Finite-time Convergence*," where a main concept precedes a subtitle. Such two-part structures may make titles

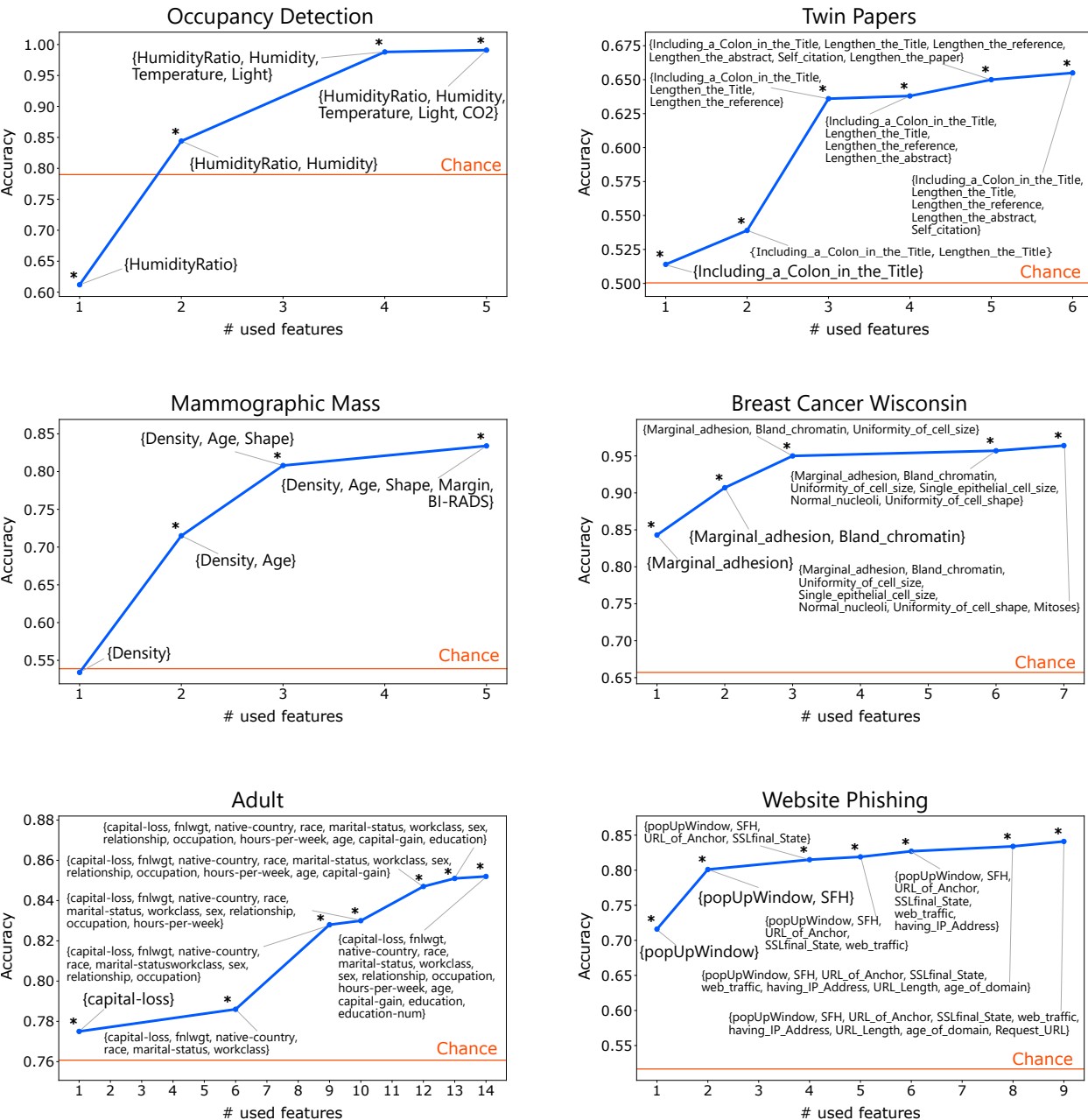

Figure 2: Test accuracy as a function of the number of interesting features $K$. The chance rate represents the accuracy obtained by always predicting the majority class. * indicates statistical significance at the $\alpha = 0.05$ level, based on a likelihood-ratio test against an intercept-only null model, with Bonferroni correction applied.

catchier and more memorable, and indeed we observe a small but consistent effect. Although an accuracy of 52% may appear to reflect weak predictive power, the test set contains as many as 17,000 instances, and the result of outperforming the chance rate by 2% represents a statistically robust signal. The result—"titles with colons are slightly more likely to be cited"—is far more non-obvious and interesting than the plain rule "papers with more references get more citations."

For comparison, one of the least interesting features according to our ranking is `Self_citation`, indicating whether a paper cites the author's own prior work. That self-citation increases total citations is obvious,

and indeed the fitted classifier assigns it a positive effect. This contrast highlights that EUREKA actively avoids trivialities and instead uncovers rules that are modest in predictive power but rich in interestingness.

Across datasets, EUREKA's rules are interpretable, communicable, and thought-provoking: humidity as a proxy for occupancy, punctuation affecting citation counts, counter-intuitive links between losses and income. Such rules spark hypotheses that would not emerge from standard accuracy-driven pipelines.

# 4 Related Work

## 4.1 Interpretability and Explainability in Machine Learning

The machine-learning community has long recognized that performance metrics alone are insufficient for evaluating real-world utility, since accuracy does not reveal the reasons behind predictions or undesired behaviours [8, 12, 26]. Because explanations are meant for humans, insights from psychology and social science are important [33]. Methods to improve interpretability include surrogate and rule-based explanations [44, 45], feature-attribution techniques [31, 50, 55], visualizations for deep neural networks [1, 3, 48, 51], counterfactual explanations [34, 60], and concept-based explanations [23]. Finally, inherently interpretable "glass-box" models such as linear regression, decision trees, and generalized additive models remain valuable because their internal mechanisms are directly understandable.

Existing literature has primarily focused on applying explanation methods to accurate models (such as deep neural networks), or on developing interpretable models that also achieve high accuracy. By contrast, our method also requires interpretability, but its central objective is interestingness rather than accuracy. We argue that interpretability is a necessary condition for identifying interesting features—an uninterpretable one cannot be deemed interesting—but not sufficient, since unexpectedness and its implications must also be considered.

## 4.2 Hypothesis Generation and AI for Science

Automated hypothesis generation began with *literature-based discovery* (LBD), which seeks "undiscovered public knowledge" by linking disjoint literatures [52, 56]. Early systems operationalized this idea via co-occurring "B-terms" and diffusion over literature graphs [53, 54], complemented by topic-model search and controlled vocabularies to scale across biomedical corpora [57].

Modern work leverages large language models and retrieval [5, 15]. Prompted or iterative pipelines refine hypotheses and support long-context reasoning [7, 61, 67]. Multi-agent and retrieval-augmented variants ground proposals in literature and knowledge graphs to improve factuality [22, 62, 63, 64]. A complementary approach uses grounding and evaluation through literature [28, 39], uses language models as a simulator [66], and employs sparse-representation analysis to exploit latent factors [35].

In contrast to these lines—which largely optimize accuracy, feasibility, or faithfulness of hypotheses—the *interestingness-first classifier* treats interestingness as the primary objective during model construction. Several concurrent studies, such as InterFeat [38] and HypoBench [27], also consider the interestingness of hypotheses. However, in those works, interestingness is treated merely as one evaluation criterion rather than as the primary objective, which differentiates our study. Furthermore, while most research on hypothesis generation with LLMs aims to produce hypotheses expressed in natural language [15, 29, 67], our work instead focuses on constructing interpretable classifiers, setting it apart from the approaches in the literature.

## 4.3 Pairwise Comparisons and Ranking Estimation

The problem of inferring global rankings from pairwise comparisons has been studied for nearly a century [6, 58]. Classical statistical models assume each item has a latent strength and posit a probabilistic model for comparison outcomes: the Thurstone-Mosteller model employs a probit link [58], while the Bradley-Terry-Luce (BTL) model uses a logit link [4, 30].

Recent research has emphasized computational and statistical efficiency under noisy comparisons [2, 36, 37]. Heckel *et al.* studied approximate ranking and top-$k$ recovery, proposing adaptive sampling schemes with

near-optimal sample complexity [18]. Shah and Wainwright analyzed the simple counting method, showing it achieves information-theoretic bounds without parametric assumptions [49]. These works form a broad foundation for pairwise comparison and ranking, motivating the methodological developments in this paper.

## 5 Limitations

One limitation of EUREKA is that it does not account for interactions among features. Some interesting rules may only emerge when multiple features are combined. The simplest way to address this limitation would be to introduce interaction features in the preprocessing.

Another promising direction is to generate multiple candidate classifiers that appear potentially interesting, and then apply an LLM to produce an interestingness ranking or top-$K$ selection of the classifiers. This approach not only increases the likelihood of obtaining more interesting classifiers, but also enables selection based on the structure of the classifiers themselves and the combinations of features they employ. Since EUREKA can easily generate multiple candidates through different random seeds or classifier choices, this strategy could be adopted without difficulty. In this study, we found that even the simple version of EUREKA was sufficient to discover interesting rules, and thus we did not proceed to this more advanced strategy to keep the proposal simple and clear. Nevertheless, for more challenging problems, this avenue appears promising.

Another limitation of EUREKA is that it cannot be applied when column names carry no semantic meaning. In some datasets, features are labeled only as "feature A," "feature B," and so on, making them unusable in this framework. A potential direction to address this issue is to leverage LLM-based techniques that can assign meaningful names to feature or embedding dimensions [40].

An inherent limitation of this study is that interestingness is a subjective notion. However, we regard this subjectivity as essential. While objective measures such as information-theoretic surprisal could serve as proxies for interestingness [11], they do not fully capture what humans actually find interesting. In our work, we deliberately refrain from relying on such objective metrics and instead emphasize subjective interestingness of classifiers—this, in our view, represents the true spirit of the interestingness-first approach.

Finally, we note the possibility of spurious correlations. This is the same limitation faced by conventional classifiers, and if necessary, causal analysis should be conducted after obtaining a classifier. That said, we believe spurious correlations are not a major concern in our problem setting. For example, the well-known rule that "Nicolas Cage appears in many movies → more people drown in swimming pools" is entirely coincidental [59], yet this rule is still interesting and in fact has been widely used and cited precisely because of its peculiarity. We regard the discovery of such spurious patterns as also valuable within our interestingness-first framework.

## 6 Conclusion

In this paper, we introduced the notion of *interestingness-first classifiers*, a new perspective on supervised learning that prioritizes surprising and non-obvious patterns over pure predictive accuracy. Through experiments on diverse benchmark datasets, we demonstrated that EUREKA systematically identifies features that deviate from the obvious yet still achieve non-trivial accuracy. For instance, EUREKA found the following classification rules.

- "High Humidity → Room Occupied" in the Occupancy Detection Dataset.

- "Title Contains Colon → Paper More Cited" in the Twin Papers Dataset.

- "More Capital *Loss* → *High* Income > \$50k" in the Adult Dataset.

It is worth emphasizing that these interesting rules were discovered entirely automatically by EUREKA, simply by feeding the dataset. These findings provide supporting evidence for the validity of the interestingness-first approach.

We hope our study inspires readers to look beyond accuracy and to also value interestingness in classifiers.

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
