# OpenReview forum: "Interestingness First Classifiers"
_TMLR — Rejected by TMLR_

### Review · Reviewer_N8SC · 2025-09-15

**Summary Of Contributions:**

The paper proposes EUREKA, a lightweight “interestingness-first” classifier. It (i) uses an LLM to perform pairwise comparisons over feature names and aggregates them (Borda) into a ranking. The authors also studied different type of ranking methodologies and provide evidence supporting their approach. (ii) Then it trains a simple, interpretable model (e.g., logistic regression) on the top-K ranked features, and (iii) increases K until performance is acceptable. The goal is to surface features that are surprising yet interpretable, rather than purely optimizing accuracy.

In this work, the authors perform experiments with 6 datasets and compare EUREKA with three baseline classifiers, Group LASSO, Logistic Regression, and Validation-based selection. The authors claim that EUREKA can find surprising and interesting features, which could not be achieved by traditional methods with accuracy-driven approaches.

**Audience:**

Yes

**Audience Explanation:**

To the best of my knowledge in the related works, this work is one of the first to touch on "finding interesting features" in prediction tasks, and this is potentially valuable for the hypothesis generation and AI for scientific discovery community. I think this work is certainly interesting and thought-provoking, and could be a valid starting point for future works in this direction.

**Claims And Evidence:**

No

**Claims Explanation:**

I do like the motivation and the framing of this work. However, there isn’t enough detail or significant evidence to support the central claim that EUREKA selects “more interesting” features, for several reasons (summarizing my prior comments):

**Compound features.** The paper studies only simple (single) features. In practical prediction tasks, important signals are often compositional (interactions, ratios, thresholds). Evaluating interestingness solely at the single-feature level is too restrictive and risks missing the genuinely informative and essentially more “interesting” structure.

**Datasets/experiment detail.** There isn’t enough information about the datasets to ground the claims: whether they’re real or synthetic datasets, how are they collected, basic statistics, and whether prior work has identified “interesting yet predictive” features on these tasks. The paper also argues that many classifiers focus only on predictive features without exploring interesting ones, but for some applications high predictive power is the main goal. With the current datasets and reporting, the motivation for prioritizing “interestingness” over accuracy is under-justified. It would be better to see that EUREKA can push the Pareto boundary of these tasks further.

**Scales in figure 2.** The accuracy plots use different y-axis scales across panels, which is visually misleading. The qualitative takeaway (“more features → better”) is fine, but the margins appear larger than they are (especially for Twin Papers). Also, the first step—adding the single “most interesting” feature—often does not produce accuracy clearly above chance; only after adding more features do gains emerge. That weakens the claim that the top “interesting” features are themselves helpful for prediction.

**Definition of interestingness.** The paper does not provide a substantive definition. I agree this is highly non-trivial, but at minimum a more structured framing or a small human study to validate the LLM-based judgments would help. Right now, interestingness reduces to subjective LLM comparisons over names, with no external validation.

Finally, I acknowledge that the authors mention some of these issues in the limitations section. Even so, several of them (interactions, dataset transparency, and a clearer/validated notion of interestingness) feel crucial to substantiate the main claim that EUREKA surfaces more interesting features.

**Requested Changes:**

Please see above. Overall, I think the aforementioned, or at least some, limitations need to be addressed to make this paper stronger.

---

> ### Author Response · Authors · 2025-09-30
>
> We thank the reviewer for the thoughtful comments. We respond point-by-point.
>
> > Compound features. The paper studies only simple (single) features. In practical prediction tasks, important signals are often compositional (interactions, ratios, thresholds). Evaluating interestingness solely at the single-feature level is too restrictive and risks missing the genuinely informative and essentially more “interesting” structure.
>
> We agree that considering multi-feature structure is important. EUREKA is applicable to composite features; in fact, some features used in our experiments (e.g., HumidityRatio in Occupancy) are already composite.
>
> ### New experiment:
>
> We added composite features (ratio and product) and re-ran EUREKA for the Occupancy dataset.
>
> Results:
>
> The top feature selected by EUREKA was `Temperature / HumidityRatio` (ratio), and the second feature was `Temperature * HumidityRatio` (product).
>
>
> | Feature set                                                   | Test Accuracy |
> | ------------------------------------------------------------- | ------------- |
> | Chance (majority class)                                       | 0.79          |
> | `Temperature / HumidityRatio`                                 | 0.736         |
> | `Temperature / HumidityRatio` + `Temperature * HumidityRatio` | 0.819         |
>
> As before, obvious features such as Light were not selected by EUREKA. The first feature alone was not predictive, but the two interesting, non-obvious composites together achieved non-trivial accuracy above chance.
>
> > Datasets/experiment detail. There isn't enough information about the datasets to ground the claims: whether they're real or synthetic datasets, how are they collected, basic statistics, and whether prior work has identified “interesting yet predictive” features on these tasks. The paper also argues that many classifiers focus only on predictive features without exploring interesting ones, but for some applications high predictive power is the main goal. With the current datasets and reporting, the motivation for prioritizing “interestingness” over accuracy is under-justified. It would be better to see that EUREKA can push the Pareto boundary of these tasks further.
>
> We appreciate the suggestion to frame that EUREKA can push the Pareto boundary; this basically matches our intent. We focused on the extreme case of the Pareto boundry, i.e., interesingness first, in this paper but we agree that the entire Pareto boundary is also important.
>
> All six datasets are real, publicly available benchmarks. The descriptions are as follows. We include them in the camera-ready.
>
> | Dataset                  | Collection Method                                                                 | Basic Statistics                |
> | ------------------------- | -------------------------------------------------------------------------------- | -------------------------------- |
> | Occupancy Detection       | Office sensors (temperature, humidity, light, CO2) with ground-truth from photos | 19,987 instances, 5 features     |
> | Twin Papers               | Extracted from AMiner; pairs of similar academic papers published in parallel    | 87,396 instances, 7 features     |
> | Mammographic Mass         | Clinical records from 961 patients (University of Erlangen-Nuremberg, 2003-2006) | 961 instances, 5 features        |
> | Breast Cancer Wisconsin   | University of Wisconsin Hospitals records (1989-1991)                            | 699 instances, 9 features        |
> | Adult                     | Extracted from 1994 US Census; filtered to working adults                        | 48,842 instances, 14 attributes  |
> | Website Phishing          | URLs from PhishTank (phishing) + crawled from Yahoo (legitimate)                 | 1,353 instances, 9 features      |
>
> > Scales in figure 2. The accuracy plots use different y-axis scales across panels, which is visually misleading. The qualitative takeaway (“more features → better”) is fine, but the margins appear larger than they are (especially for Twin Papers). Also, the first step—adding the single “most interesting” feature—often does not produce accuracy clearly above chance; only after adding more features do gains emerge. That weakens the claim that the top “interesting” features are themselves helpful for prediction.
>
> Thank you for point out the issue. We will fix it in the camera ready.

---

> > ### Comment · Reviewer_N8SC · 2025-10-14
> >
> > Thank you for the follow-up details.
> >
> > While these are helpful, I still think the major concerns still hold, especially the missing definition of interestingness. I think focusing on the extreme case of the pareto boundary on interestingness is a valid approach, but almost all experiment results are about prediction accuracy, and there is little concrete evidence that EUREKA indeed generates more interesting predictors. In order to show this, having some measurements on interestingness and compare with more recent methods on generating interesting predictors could be helpful.

---

> > > ### Author Response · Authors · 2025-10-15
> > >
> > > We thank the reviewer again for the continued engagement and helpful clarifications.
> > >
> > > > While these are helpful, I still think the major concerns still hold, especially the missing definition of interestingness. I think focusing on the extreme case of the pareto boundary on interestingness is a valid approach, but almost all experiment results are about prediction accuracy, and there is little concrete evidence that EUREKA indeed generates more interesting predictors. In order to show this, having some measurements on interestingness and compare with more recent methods on generating interesting predictors could be helpful.
> > >
> > > As stated in Section 5 (Limitation), the notion of interestingness in this work is subjective, but we consider this subjectivity to be essential. The reviewer requests a definition of interestingness, but we believe that such a definition is fundamentally impossible. Even if one could establish an objective criterion for "interestingness" and build a classifier that "maximizes" it, that very criterion, once turned into an objective numerical measure, would cease to be interesting precisely because it is an objective value, or at least there would inevitably be people who feel that way.
> > > For this reason, we deliberately refrained from relying on any objective metrics. Our contribution lies precisely in acknowledging this essential limitation and still turning our attention toward subjective interestingness rather than predictive accuracy.
> > >
> > > Indeed some people may find EUREKA's ordering not interesting, but what we intend is that some people may find EUREKA's ordering interesting, and others may not. Those who do not can simply choose not to use it, while those who do can make use of it. For those whose notion of interestingness aligns with EUREKA, it can reduce the time needed to reach promising rules by suggesting an ordering to check first, rather than inspecting the full feature list manually. Eureka can provide an alternative ordering rather than predictive accuracy.

---

> > > > ### Comment · Reviewer_N8SC · 2025-10-16
> > > >
> > > > I agree to the point that a definition of interestingness is fundamentally difficult, and I do think this is valuable effort. From a practical perspective, it would be better to have other confirmations or evidence of EUREKA generates interesting and preferrably more interesting features than those baselines from outside the author group.

---

> ### Author Response · Authors · 2025-10-20
>
> Thank you for this valuable suggestion. To address the reviewer's concern, we conducted a real-world human evaluation with 100 independent crowd workers to compare the interestingness of rules generated by EUREKA against those from accuracy-first baselines.
>
> For each task, participants were asked which rule they found more "interesting":
>
> ## Room occupancy prediction
> High Humidity → Room Occupied (EUREKA's rule)
> vs.
> Lights On → Room Occupied (accuracy-first rule)
>
> ## Citation prediction
> Title Contains Colon → Paper More Cited (EUREKA's rule)
> vs.
> Long Reference List → Paper More Cited (accuracy-first rule)
>
> ## Income prediction
> More Capital Loss → High Income (EUREKA's rule)
> vs.
> More Capital Gain → High Income (accuracy-first rule)
>
> ## Results
>
> | Rule                                              |  Votes |
> | :------------------------------------------------ | -----: |
> | High Humidity → Room Occupied (**EUREKA’s rule**) | **86** |
> | Lights On → Room Occupied (accuracy-first rule)              |     14 |
>
> | Rule                                                        |  Votes |
> | :---------------------------------------------------------- | -----: |
> | Title Contains Colon → Paper More Cited (**EUREKA’s rule**) | **68** |
> | Long Reference List → Paper More Cited (accuracy-first rule)           |     32 |
>
> | Rule                                                          |  Votes |
> | :------------------------------------------------------------ | -----: |
> | More Capital Loss → High Income (**EUREKA’s rule**) | **84** |
> | More Capital Gain → High Income (accuracy-first rule)          |     16 |
>
>
> These results support that EUREKA systematically produces rules perceived as more interesting by independent human evaluators, providing external confirmation beyond the author group.
>
> We hope these results resolve your concern.

---

### Review · Reviewer_dUsp · 2025-09-16

**Summary Of Contributions:**

The paper introduces EUREKA, a framework for building classifiers that prioritize interestingness over predictive accuracy. The method ranks feature names using LLM-based pairwise comparisons of their “interestingness” and then trains interpretable models on the top-ranked features. The paper demonstrates the approach on various datasets, yielding non-obvious decision rules such as “humidity predicts occupancy” (rather than the usual prediction based on CO2 levels)

The paper is well structured and the shift from accuracy to interestingness is certainly novel, adding illustrative examples along the way.
However, “interestingness” is left undefined beyond LLM judgments without a methodical operationalization or human expert validation. In addition, the LLM judgment relies uniquely on the feature name, which makes its justification too weak. The application of this technique to a dataset of a more obscure domain might be useless, as (1) the LLM might not possess information about it and (2) traditional rule induction (based on accuracy) might already help towards knowledge discovery.

**Additional Comments:**

-

**Audience:**

Yes

**Audience Explanation:**

The idea of optimizing classifiers for interestingness rather than accuracy is certainly curious and could generate discussion among researchers in XAI. However, there is clear room for improvement, and we believe this work is better suited for a small conference or workshop and should be rejected for this specific journal. We encourage the author to do this, as the framing challenges conventional evaluation metrics and may inspire new directions.

**Broader Impact Concerns:**

-

**Claims And Evidence:**

No

**Claims Explanation:**

While the paper presents experimental evidence that the method can produce interpretable, non-obvious rules, it does not convincingly demonstrate that these rules are interesting in a reproducible or validated sense, with the overreliance on LLM judgments undermining the credibility of the main claim.

Tying the notion of interestingness to only the feature name rather than to the entirety of the data itself is rather lackluster and too subjective. What if two datasets with the same feature have very different data distributions?

Furthermore, the method presented is essentially a filter feature subset selection algorithm that focuses on “interestingness” rather than predictive performance. Contextualization and evaluation from the perspective of this framework are lacking. Maybe the author can use this book for reference: https://www.sciencedirect.com/science/article/pii/S016794731930194X

Overall, the evaluation lacks depth.

**Requested Changes:**

Provide an operational proposal of “interestingness,” beyond LLM subjectivity. Perhaps non-linear feature-target effects? Local accuracy? Or at least leverage only partially on LLMs judgement.

Another suggestion for interestingness would be to study the feature importance and/or feature effects of a foundation model, such as TabPFN (where several explainability methods have been efficiently implemented)  and extract the notion of interestingness from here. Perhaps a feature that performs well in a local/fine-tuned model but that is average in TabFPN could be considered interesting.

If the authors decide that a formalization is not possible (or desirable), validate results with human expert evaluations to confirm that the selected rules are perceived as interesting. This would, however, require a more domain-specific, informal and empirical definition of interestingness.

Benchmarking this work through the lens of feature subset selection (specifically filter-based methods) would be an addition that we believe will certainly benefit this work.

Some future directions pointed out by the author himself could positively impact this work without being too distracting: feature effects, causal analysis…

---

> ### Author Response · Authors · 2025-09-30
>
> We thank the reviewer for the constructive comments. Below we address the main points in order.
>
> > While the paper presents experimental evidence that the method can produce interpretable, non-obvious rules, it does not convincingly demonstrate that these rules are interesting in a reproducible or validated sense, with the overreliance on LLM judgments undermining the credibility of the main claim.
>
> We do not claim that EUREKA by itself produces perfect classifiers. The contribution is that it offers a fully automated and low-cost way to generate candidate classifiers that are potentially interesting. We agree that leaving the final decision entirely to LLM judgments would be risky. Our intended use is that if a rule discovered by EUREKA is interesting for the user, they may adopt it; if not, they may discard it or generate another rule. (In scientific contexts, repeated exploration would naturally require proper statistical controls such as p-value adjustment, though.) In all cases, the final judgment rests with the human user, while EUREKA provides an automated way to surface candidates that may help discover interesting rules.
>
> > Tying the notion of interestingness to only the feature name rather than to the entirety of the data itself is rather lackluster and too subjective. What if two datasets with the same feature have very different data distributions?
>
> Our main goal is to produce interesting classifiers in a fully automatic and low-cost fashion, and we believe that feature names are a good starting point for this. We acknowledge the possibility of false positives, but we prioritized simplicity, as we discussed in Section 5. In our experiments, we showed that feature names alone already yield rules that are distinguishable and predictive to some degree. Of course, one could extend the framework to incorporate information about data distributions, but doing so would add bells and whistles, increasing complexity, cost, and management overhead. We deliberately chose the simplest possible design of using feature names to make the essence of the method clear.
>
> > Another suggestion for interestingness would be to study the feature importance and/or feature effects of a foundation model, such as TabPFN (where several explainability methods have been efficiently implemented) and extract the notion of interestingness from here. Perhaps a feature that performs well in a local/fine-tuned model but that is average in TabFPN could be considered interesting.
>
> We appreciate this suggestion. It is indeed an interesting direction to define interestingness. While this goes beyond the present scope, we will add it as a promising future direction in the camera-ready.
>
> > Benchmarking this work through the lens of feature subset selection (specifically filter-based methods) would be an addition that we believe will certainly benefit this work.
>
> As you noted, EUREKA can be seen as a kind of filter method, but with the key distinction that the filtering criterion is interestingness rather than predictive power. Naturally, in terms of accuracy, existing filter methods will outperform EUREKA. At the same time, EUREKA is the only method that filters explicitly by interestingness. We will add a paragraph in the Related Work and Discussion, referencing and discussing the literature you suggested and representative filter methods such as those based on F-score or mutual information in the camera ready.

---

> > ### Comment · Reviewer_dUsp · 2025-10-21
> >
> > Thanks for engaging with the comments and for updating the PDF. In general, I think the paper was greatly improved. Below some remarks:
> >
> > > they may discard it or generate another rule. (In scientific contexts, repeated exploration would naturally require proper statistical controls such as p-value adjustment, though.)
> >
> > This is a central issue, I believe. If a scientific experiment (say, filter feature subset selection) is repeated, either (1) some conclusion can be extracted about what to tweak if the result is not satisfactory (increase or decrease parameter, change a metric...) or (2) at least a procedural search can be carried out. I am unsure this is possible with EUREKA, where the whole pipeline relies on "random" feature subset selection and then the user deciding if it is good enough.
> >
> > >  Our intended use is that if a rule discovered by EUREKA is interesting for the user, they may adopt it; if not, they may discard it or generate another rule.
> >
> > I appreciate the small experiment as a result with N8SC showing that rules by EUREKA is preferred, the claims are certainly better justified. Larger experimentation in other domains might still be necessary for publication in a top tier venue, but this is still a promising result
> >
> > > We will add a paragraph in the Related Work and Discussion, referencing and discussing the literature you suggested and representative filter methods such as those based on F-score or mutual information in the camera ready.
> >
> > I believe a larger highlight might be needed, as this is by definition (or at the very least 90% like) a filter method (a subset of variables is selected prior to training a classifier). Perhaps not a full experiment, but at least framing it inside the field of (filter) feature subset selection.

---

> > > ### Author Response · Authors · 2025-10-23
> > >
> > > Thank you for the thoughtful comments and for recognizing the improvement of the paper.
> > >
> > > > This is a central issue, I believe. If a scientific experiment (say, filter feature subset selection) is repeated, either (1) some conclusion can be extracted about what to tweak if the result is not satisfactory (increase or decrease parameter, change a metric...) or (2) at least a procedural search can be carried out. I am unsure this is possible with EUREKA, where the whole pipeline relies on "random" feature subset selection and then the user deciding if it is good enough.
> > >
> > > This is possible with EUREKA with appropriate p-value correction. This is common with the standard procedure for iterative experiments. In our experiments, we applied Bonferroni correction, and even after this adjustment, the classifiers generated by EUREKA remained statistically significant. Of course, too many retries would decrease statistical power; however, (i) in most cases, simpler hypotheses are preferred, so one would only examine the top-K discovered rules with K kept small, and small K leads to a small correction penalty, and (ii) as discussed in the additional experiments with crowd workers, EUREKA tends to produce rules that are more likely to be perceived as "interesting," thus requiring fewer exploratory iterations to reach preferable rules compared to other approaches, which further mitigates the need for heavy correction.
> > >
> > > We also appreciate your point about situating our method within the context of filter-based feature subset selection. We will revise the Related Work section in the camera ready to more clearly frame EUREKA within the field of filter-based feature subset selection.

---

### Review · Reviewer_jVyk · 2025-09-27

**Summary Of Contributions:**

The paper introduces a method that selects features according to their "interestingness" as evaluated by an LLM. These features are then used to build light-weighted classifiers which prioritize interestingness over accuracy. While the idea might sound curious at first, it comes with a number of serious concerns that unfold throughout the paper. In summary, it is unclear to me (1) what the purpose of the interesting-first classier might be, (2) how its insights differ from any other classifier when considering the full feature set, and (3) how the fundamental issue of surfacing spurious correlations could be solved in this context.

**Audience:**

Yes

**Audience Explanation:**

The paper appears to introduce a creative idea which might spur initial interest.

**Broader Impact Concerns:**

My third concern in the list above would warrant adding a statement about ethical implications.

**Claims And Evidence:**

No

**Claims Explanation:**

I have a number of concerns, some of them being rather fundamental:

- First, what is the envisioned use case of the proposed method? Since its not about building accurate models, certainly the models should not be used to inform, e.g., decision-making, especially not in high-risk settings like in the medical domain. In my view its also not positioned for informing substantive theory since this can be done with existing models (see next point).

- Second, what is the benefit of the present approach over any other standard SML method? The paper almost seems to pretend that current methods only select one single predictive feature. Of course any classifier allows researchers to rank features and explore the larger list in search for "interesting" features or unexpected interactions, besides the top predictive ones. What EUREKA does seems to be just turning that feature list upside down, which in my view is not a significant achievement.

- Third, the selected "interesting" features can provoke misleading conclusions. My biggest concern is that EUREKA might select confounder as "interesting" features, although they only show spurious correlations with the outcome and in fact should not be relied upon. I believe this can be quite problematic in sensitive domains, and certainly not "valuable" as claimed in the discussion. This directly relates to the fact that the paper uses cancer prediction tasks in the experiments, which I feel are misplaced here as perceived "interestingness" in such contexts should be judged by domain experts in my view.

- Fourth, the paper uses benchmark data which have been criticized for various reasons (Ding et al. 2021, Fabris et al. 2022). What can be gained from a discussion of results from data collected in 1994 (Adult)? Is it really "interesting" that capital loss is also a predictor of high income, next to capital gain? Is it "interesting" that the model picks features such as race, native country, sex and marital status, which are considered protected attributes under anti-discrimination law?


Ding, F., Hardt, M., Miller, J., and Schmidt, L. (2021). Retiring adult: new datasets for fair machine learning. In Proceedings of the 35th International Conference on Neural Information Processing Systems (NIPS '21). Curran Associates Inc., Red Hook, NY, USA, Article 496, 6478–6490.

Fabris, A., Messina, S., Silvello, G. et al. (2022). Algorithmic fairness datasets: the story so far. Data Min Knowl Disc 36, 2074–2152. https://doi.org/10.1007/s10618-022-00854-z

**Requested Changes:**

I believe the fundamental assumption behind the paper should be questioned and revised. Models cannot do the theoretical work that decades of substantive research in the respective application domains have done, and simply querying an LLM about what might be interesting features is not a sensible substitution or addition to domain expertise.

---

> ### Author Response · Authors · 2025-09-30
>
> We thank the reviewer for the careful reading and constructive critique. We address the four main concerns point-by-point, clarify scope and claims, and report an additional experiment on a modern dataset (Folktables; CA 2018). We will incorporate the proposed clarifications and edits in the camera-ready.
>
> > First, what is the envisioned use case of the proposed method? Since its not about building accurate models, certainly the models should not be used to inform, e.g., decision-making, especially not in high-risk settings like in the medical domain. In my view its also not positioned for informing substantive theory since this can be done with existing models (see next point).
>
> We do not intend EUREKA for deployment in high-risk domains (e.g., medicine). If used at all in such areas, the right place is early-stage hypothesis exploration where the goal is to inspire investigators, not to inform decisions. As we note in Section 5, follow-up causal analyses and expert review are needed before any substantive interpretation in sensitive settings. Even so, we believe EUREKA helps before formal hypothesis testing by surfacing non-obvious candidate rules that can be evaluated next.
>
> Beyond high-risk domains, we envision low-risk applications. For instance, if humidity alone gives a usable proxy for room occupancy, a hobbyist might adapt a hygrometer to monitor a study room without installing CO2 sensors. Likewise, if adding a colon to a paper title is associated with higher citations, that may inform title-writing practices. More broadly, imaginative users can find many ways to make use of such lightweight heuristics when perfect accuracy is not required.
>
> Finally, there is a clear educational use case. The well-known spurious rule "Nicolas Cage appears in many movies → more people drown in swimming pools" is cited precisely because it is memorable and interesting. Finding such examples by hand is costly; EUREKA may help surface them automatically. Our "colon-in-title → higher future citations" rule has indeed a causal effect and can be an inspiring example in a textbook on causal inference with its nuance.
>
>
> > Second, what is the benefit of the present approach over any other standard SML method? The paper almost seems to pretend that current methods only select one single predictive feature. Of course any classifier allows researchers to rank features and explore the larger list in search for "interesting" features or unexpected interactions, besides the top predictive ones. What EUREKA does seems to be just turning that feature list upside down, which in my view is not a significant achievement.
>
> We agree that, in principle, one could manually explore long feature lists from standard models and eventually encounter something interesting, but this is labor intensive. Our contribution is that EUREKA does this end-to-end automatically. That is different from simply "flipping" a predictiveness ranking. The method hinges on LLM-based pairwise judgments aggregated by Borda counts to produce a stable interestingness order, and even the variants including rank-all-at-once prompting have different results and the proposed method is more stable (Table 1), and competitive with an active-ranking baseline while being easier to parallelize (Fig. 1). Across datasets, EUREKA does not simply pick the least predictive variables.

---

> ### Author Response · Authors · 2025-09-30
>
> > Third, the selected "interesting" features can provoke misleading conclusions. My biggest concern is that EUREKA might select confounder as "interesting" features, although they only show spurious correlations with the outcome and in fact should not be relied upon. I believe this can be quite problematic in sensitive domains, and certainly not "valuable" as claimed in the discussion. This directly relates to the fact that the paper uses cancer prediction tasks in the experiments, which I feel are misplaced here as perceived "interestingness" in such contexts should be judged by domain experts in my view.
>
> We agree that "interesting" features can include confounders and may be spurious. For our aims, this is not a defect: EUREKA is a candidate-rule generator, not a certifier of causal truth. We reiterate that we do not propose deployment in sensitive settings; in educational contexts, confounding itself can be the point (e.g., the Nicolas Cage example), precisely because it sparks discussion about correlation vs. causation. We also agree that, in specialized domains, domain experts should judge interestingness and filter candidates; the value of EUREKA is to screen at low cost and prioritize which patterns merit expert attention. We will make these guardrails more explicit by expanding Sec. 5 (Limitations) to state, no causal claims and expert review required in sensitive domains, in the camera-ready.
>
>
> > Fourth, the paper uses benchmark data which have been criticized for various reasons (Ding et al. 2021, Fabris et al. 2022). What can be gained from a discussion of results from data collected in 1994 (Adult)? Is it really "interesting" that capital loss is also a predictor of high income, next to capital gain? Is it "interesting" that the model picks features such as race, native country, sex and marital status, which are considered protected attributes under anti-discrimination law?
>
> Our goal is not to build a fair or deployable classifier; therefore using familiar benchmarks (including Adult) is acceptable for mechanism-level illustration. If one needs fairness, there are may specialized methods to impose it; our aim is different. We also maintain that the finding "capital loss relates to high income" is interesting in its counter-intuitive flavor and invites explanations, as discussed in Section 3.2–3.3. We do not endorse deploying rules that use protected attributes; rather, it is educationally illuminating that, absent fairness constraints, protected attributes can deliver strong predictiveness and that this is exactly why such features must be handled with care. We also agree that what counts as interesting is inherently subjective; appreciating the interestingness of these rules may require taste and sensibility.
>
> In response to the reviewer's point and the critiques of Adult in the literature, we additionally ran EUREKA on Folktables (CA 2018). Note that capital gain/loss are not present in this dataset. The task is income prediction. EUREKA’s ranking and accuracy:
>
> ```
>  Most interesting: Marital status
>  Second: Place of birth
>  ...
>  Second least interesting: Educational attainment
>  Least interesting: Occupation
> ```
>
> ### Accuracy:
>
> | Feature set                             | Accuracy |
> | --------------------------------------- | -------: |
> | Chance (majority)                       |     58.9 |
> | Top-1 (`Marital status`)                  |     61.4 |
> | Top-2 (`Marital status` + `Place of birth`) |     65.0 |
> | All features                            |     78.6 |
>
> We find it more interesting and consistent with our objective that non-obvious sociological attributes (`Marital status`; `Place of birth`) deliver non-trivial accuracy above chance, compared with relying on `Educational attainment` or `Occupation`, which are obvious predictors.
>
> In sum, our scope is exploration, inspiration, and education, not deployment in high risk domains. EUREKA’s contribution is to automate the surfacing of interesting rules and to show that such rules can still surpass chance. We will update the manuscript accordingly with clarifications and the Folktables experiments in the camera ready.

---

> > ### Comment · Reviewer_jVyk · 2025-09-30
> > **thanks & brief follow-up**
> >
> > Thank you for engaging with my comments. While the outlined explanations and guardrails would indeed be important to add, let me briefly follow-up regarding the intended use case and potential risks.
> >
> > In research applications, we could envision two scenarios:
> >
> > - 1. A researcher with domain expertise applies EUREKA: This is where I'm unsure what the added benefit over manual inspection of the feature list would be since the list can easily be filtered to predictors that rank above a pre-specified performance level. Then, I would argue that the domain expert would be the only right instance to judge "interestingness", and whether or not they might agree with EUREKA is a totally open question.
> >
> > - 2. A researcher without domain expertise applies EUREKA: This is where my third comment on the risk of misleading conclusion applies, since any confounder or proxy variable (see below) that EUREKA surfaces might be taken at face value. To prevent such (mis)use, in my view the paper should be framed very differently and not present EUREKA as a general purpose method that can "just" be applied to any prediction problem in complex domains such as medicine or sociology as suggested by the experiments.
> >
> > A note on the adult example: I understand that the paper is not proposing a bias mitigation method; my comment was meant to illustrate that EUREKA can surface predictors which can lead to misleading conclusions if they are not contextualized. I would argue that features such as race or marital status are proxy variables whose effect on income can be explained by group-specific differences in a range of socio-economic factors and circumstances, next to direct discrimination. This has long been studied in sociology, which brings me back to my argument above: For research applications, the role of EUREKA vis-a-vis substantive domain knowledge is unclear to me as the latter cannot be automated or outsourced to an LLM.

---

> > > ### Author Response · Authors · 2025-10-02
> > >
> > > We thank the reviewer again for the continued engagement and helpful clarifications.
> > >
> > > > I would argue that the domain expert would be the only right instance to judge "interestingness", and whether or not they might agree with EUREKA is a totally open question.
> > >
> > > We agree that there is no guarantee that domain experts will always find EUREKA's ranking aligned with their own sense of what is interesting. What we intend is that, some people may find EUREKA’s ordering interesting, and others may not; those who do not can simply choose not to use it, while those who do can make use of it. For those whose notion of interestingness aligns with EUREKA, it can reduce the time needed to reach promising rules by suggesting an ordering to check first, rather than inspecting the full feature list manually.
> > >
> > > > A researcher without domain expertise applies EUREKA: This is where my third comment on the risk of misleading conclusion applies, since any confounder or proxy variable (see below) that EUREKA surfaces might be taken at face value. To prevent such (mis)use, in my view the paper should be framed very differently and not present EUREKA as a general purpose method that can "just" be applied to any prediction problem in complex domains such as medicine or sociology as suggested by the experiments.
> > >
> > > We appreciate this concern. Our position is that this problem is not unique to EUREKA but is shared by virtually all feature selection or interpretability tools. For example, LASSO can select confounders and SHAP can highlight proxy variables, yet these methods are still published as general-purpose tools. Misuse is always possible, but this does not in itself disqualify a method. That said, we agree that the risk should be explicitly acknowledged. In the camera-ready we will strengthen Section 5 (Limitations) to explicitly state that EUREKA may surface confounders or proxy variables, and that in sensitive domains this possibility must be considered carefully.
> > >
> > > We hope these clarifications address the reviewer's concerns, and we would be glad to further elaborate if additional questions or points remain.

---

### Author Response · Authors · 2025-09-30

We thank all reviewers for their thoughtful and constructive feedback. We are encouraged that the framing of interestingness-first classifiers is regarded as novel and thought-provoking, and we take seriously the concerns raised about scope, validation, datasets, and presentation. Below we summarize the key clarifications, changes, and new experiments that will be incorporated in the camera-ready version.

Reviewer N8SC highlighted that important signals are often compositional. We agree this point. EUREKA can naturally incorporate composite features, and in fact some features already are (e.g., HumidityRatio).

We conducted additional experiments to strengthen this point.

### New experiment

we generated ratio and product features for the Occupancy dataset and re-ran EUREKA:

The top feature selected by EUREKA was `Temperature / HumidityRatio` (ratio), and the second feature was `Temperature * HumidityRatio` (product).

| Feature set                                                   | Test Accuracy |
| ------------------------------------------------------------- | ------------- |
| Chance (majority class)                                       | 0.79          |
| `Temperature / HumidityRatio`                                 | 0.736         |
| `Temperature / HumidityRatio` + `Temperature  HumidityRatio` | 0.819         |

As before, obvious features such as Light were not selected by EUREKA. The first feature alone was not predictive, but the two interesting, non-obvious composites together achieved non-trivial accuracy above chance.


Reviewer jVyk raised a concern on the Adult dataset. Since our aim is not to construct fair classifiers, we do not consider this as a crucial issue, but we also added an experiment with the Folktables dataset. Unlike Adult, this dataset does not contain capital gain/loss.
EUREKA’s ranking and accuracy:

```
 Most interesting: Marital status
 Second: Place of birth
 ...
 Second least interesting: Educational attainment
 Least interesting: Occupation
```

### Accuracy:

| Feature set                             | Accuracy |
| --------------------------------------- | -------: |
| Chance (majority)                       |     58.9 |
| Top-1 (`Marital status`)                  |     61.4 |
| Top-2 (`Marital status` + `Place of birth`) |     65.0 |
| All features                            |     78.6 |

We find it more interesting and consistent with our objective that non-obvious sociological attributes (`Marital status`; `Place of birth`) deliver non-trivial accuracy above chance, compared with relying on `Educational attainment` or `Occupation`, which are obvious predictors.

We include these new results in the camera ready.

We hope these clarifications and additions address the reviewers' concerns and strengthen the contribution of the paper.

---

### Author Response · Authors · 2025-10-20

Based on the discussion with Reviewer N8SC, we conducted an additional experiment. We conducted a real-world human evaluation with 100 independent crowd workers to compare the interestingness of rules generated by EUREKA against those from accuracy-first baselines.

For each task, participants were asked which rule they found more "interesting":

## Room occupancy prediction
High Humidity → Room Occupied (EUREKA's rule)
vs.
Lights On → Room Occupied (accuracy-first rule)

## Citation prediction
Title Contains Colon → Paper More Cited (EUREKA's rule)
vs.
Long Reference List → Paper More Cited (accuracy-first rule)

## Income prediction
More Capital Loss → High Income (EUREKA's rule)
vs.
More Capital Gain → High Income (accuracy-first rule)

## Results

| Rule                                              |  Votes |
| :------------------------------------------------ | -----: |
| High Humidity → Room Occupied (**EUREKA’s rule**) | **86** |
| Lights On → Room Occupied (accuracy-first rule)              |     14 |

| Rule                                                        |  Votes |
| :---------------------------------------------------------- | -----: |
| Title Contains Colon → Paper More Cited (**EUREKA’s rule**) | **68** |
| Long Reference List → Paper More Cited (accuracy-first rule)           |     32 |

| Rule                                                          |  Votes |
| :------------------------------------------------------------ | -----: |
| More Capital Loss → High Income (**EUREKA’s rule**) | **84** |
| More Capital Gain → High Income (accuracy-first rule)          |     16 |


These results quantitatively support that EUREKA systematically produces rules perceived as more interesting by human evaluators.

We hope these results address the reviewers' concerns and strengthen the contribution of the paper.

---

### Decision · Action_Editor_fYLR · 2025-10-30

**Recommendation:** Reject

**Additional Comments:**

The submission explores an unconventional and creative research direction. In its current version, however, it remains too preliminary. Authors should provide a clearer methodological grounding, strengthen validation, and more rigorously discuss risks of spurious correlations and potential misuse.

**Audience:**

Yes

**Audience Explanation:**

The idea of shifting the focus from accuracy to interestingness in feature discovery challenges a core assumption in standard machine learning evaluation and could stimulate discussion in the interpretability community.

**Claims And Evidence:**

No

**Claims Explanation:**

The current evidence doesn't quite back up the core claims convincingly. The definition of "interestingness" feels vague and is only loosely grounded in LLM-based assessments of feature names, which are inherently subjective. The human study included is a step in the right direction, but it's fairly limited. On the methodological front, EUREKA essentially operates as a filter-based feature selection method.